# MaskLLM: Learnable Semi-Structured Sparsity for Large Language Models

**Gongfan Fang**♣,◇ †  **Hongxu Yin**◇  **Saurav Muralidharan**◇  **Greg Heinrich**◇
**Jeff Pool**◇  **Jan Kautz**◇  **Pavlo Molchanov**◇‡  **Xinchao Wang**♣‡
NVIDIA◇  National University of Singapore♣
gongfan@u.nus.edu, xinchao@nus.edu.sg
{dannyy,sauravm,gheinrich,jpool,jkautz,pmolchanov}@nvidia.com

## Abstract

Large Language Models (LLMs) are distinguished by their massive parameter counts, which typically result in significant redundancy. This work introduces MaskLLM, a learnable pruning method that establishes Semi-structured (or "N:M") Sparsity in LLMs, aimed at reducing computational overhead during inference. Instead of developing a new importance criterion, MaskLLM explicitly models N:M patterns as a learnable distribution through Gumbel Softmax sampling. This approach facilitates end-to-end training on large-scale datasets and offers two notable advantages: 1) *High-quality Masks* - our method effectively scales to large datasets and learns accurate masks; 2) *Transferability* - the probabilistic modeling of mask distribution enables the transfer learning of sparsity across domains or tasks. We assessed MaskLLM using 2:4 sparsity on various LLMs, including LLaMA-2, Nemotron-4, and GPT-3, with sizes ranging from 843M to 15B parameters, and our empirical results show substantial improvements over state-of-the-art methods. For instance, leading approaches achieve a perplexity (PPL) of 10 or greater on Wikitext compared to the dense model's 5.12 PPL, but MaskLLM achieves a significantly lower 6.72 PPL solely by learning the masks with frozen weights. Furthermore, MaskLLM's learnable nature allows customized masks for lossless application of 2:4 sparsity to downstream tasks or domains. Code is available at https://github.com/NVlabs/MaskLLM.

## 1 Introduction

Large Language Models (LLMs) have demonstrated remarkable effectiveness across a diverse range of tasks [19, 6, 48, 13]. However, the generality and robustness of LLMs are largely attributed to their vast scale, with parameter counts ranging from one billion to several hundred billion [39, 45, 5]. This substantial model size, in turn, makes it challenging and resource-intensive to deploy LLMs in real-world applications. One effective and practical approach to address this issue is semi-structured pruning [29, 32, 12, 38], which introduces N:M sparsity into LLMs to improve both memory and computational efficiency. The N:M pattern, with N non-zero values

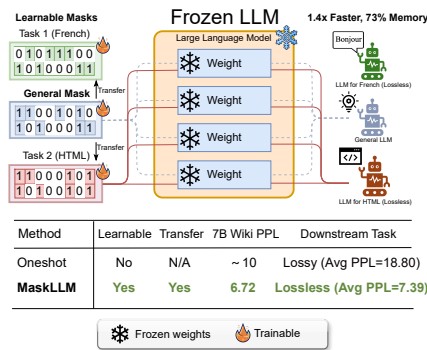

Figure 1: Learnable N:M sparsity for Large Language Models.

---

†The work was done at NVIDIA

‡Corresponding Authors: Pavlo Molchanov, Xinchao Wang

38th Conference on Neural Information Processing Systems (NeurIPS 2024).

among M consecutive parameters, is typically hardware-friendly to accelerators like GPUs and has thus garnered considerable attention [12, 38, 32].

Despite the simplicity of its core idea, semi-structured pruning still presents considerable challenges within the realm of LLMs. Sparsity aims to identify a subset of parameters that attain a comparable quality to the dense model. Nevertheless, the extensive parameter scale of large language models usually leads to a vast search space. In a fully sparsified LLaMA2-7B model with 2:4 sparsity, for instance, there are 1.6 billion 2:4 masks to be chosen for dense layers. This makes the combinatorial problem of finding the optimal mask set exceedingly challenging. In the literature, leading approaches such as SparseGPT [12] and Wanda [38], utilize a small calibration set and carefully designed importance criteria to identify redundant parameters. While these techniques have demonstrated remarkable results on several large language models, two substantial challenges remain: **Firstly, the small calibration set is insufficient to represent the comprehensive knowledge embedded in LLMs**, which are pre-trained on extensive and diverse data domains [39, 7, 31]. As demonstrated in our experiments, hand-crafted importance criteria are only applicable to a compact subset of data, and enlarging the calibration set beyond 256 entries does not improve the resulting quality. This limits the generalizability of pruned LLMs in different domains. **Secondly, using handcrafted criteria as a proxy for the true discrepancy inevitably results in errors**. A considerable gap remains between the real discrepancy induced by pruning and existing importance indicators, such as gradient information [30], weight magnitude [15], and the Hessian Matrix [23, 12].

To tackle the outlined challenges, we propose MaskLLM, a learnable method that facilitates end-to-end training of LLM sparsity on large-scale datasets. In the context of N:M sparsity, pruning an LLM involves selecting masks from a discrete and finite set. However, the non-differentiability of mask selection and combination hinders the direct use of backpropagation for mask learning. To address this, our work frames the mask selection problem from a probabilistic perspective, associating each candidate mask with a probability and modeling the mask selection as a stochastic sampling process. We incorporate the Gumbel Softmax [21] for differentiable sampling, which re-parameterizes the randomness of sampling into an independent random variable. This makes the probabilities of each mask candidate optimizable with gradient descent. During training, MaskLLM aims to learn appropriate mask distributions, from which the sampled masks can preserve the original quality of dense LLMs. The differentiable mask offers two advantages in addressing the challenges mentioned above: (1) it effectively scales to large-scale datasets, thereby preserving the rich knowledge in LLMs, and (2) the end-to-end training explicitly optimizes the language modeling loss of LLMs, which exactly measures the discrepancy induced by pruning. Furthermore, inspired by the power of transfer learning, we introduce prior masks, a simple strategy to fully leverage pre-computed masks and enable fast transfer learning of sparsity across domains and tasks as illustrated in Figure 1

To evaluate the proposed method, we conduct experiments on several LLMs including LLaMA-2 7B, LLaMA-2 13B [39], Nemotron-4 15B [31], and two in-house LLMs, GPT-3 843M and GPT-3 2B pre-trained using Megatron framework [36]. Our method can learn high-quality masks for pruning through end-to-end training on large-scale datasets. For example, compared to SparseGPT which archives a perplexity (PPL) of 10.42 on LLaMA-2 7B, our method improves the PPL to 6.72, without any update to the LLM parameters. Besides, our method facilitates the learning of domain-specific masks, which can even achieve lossless compression of LLMs on some downstream tasks or domains.

The principal contribution of this work lies in a learnable method for semi-structured pruning of LLMs. MaskLLM is designed to fully harness large-scale datasets to learn accurate masks, applicable to both general-purpose and domain-specific pruning. Additionally, the framework facilitates the transfer learning of sparsity patterns across different tasks, enabling efficient training of sparsity.

## 2   Related Works

**Pruning Large Language Models.**   Network Pruning [15, 30, 17, 18, 41] have been proven an efficient approach to compress pre-trained language models via the removal of redundant parameters. According to the granularity of pruning, existing methods can be classified into three categories: Structured Pruning [26, 43, 24], Unstructured Pruning [17, 15], and Semi-Structured Pruning [12, 38, 29, 32, 33]. Structured pruning physically eliminates substructures like attention heads [26], embeddings or depth [43] in the model, facilitating acceleration independent of specialized hardware or software infrastructure [32]. However, structured approaches typically necessitate huge retraining

efforts to recover network quality due to coarse removal of parameters [26, 43, 27, 2]. Conversely, unstructured methods aim to find a sparse model by zeroing out parameters in LLMs, which is characterized by its flexibility and minimal detrimental effect on LLMs' accuracy [12, 38, 20, 42, 44]. The acceleration of sparse models is typically impeded by the irregular nature of the resulting sparse patterns, presenting challenges in achieving computational efficiency. Positioned between structured and unstructured methods, the semi-structured approach introduces hardware-friendly patterns such as N:M sparsity, which leaves only $N$ nonzero values in each group of $M$ values and thereby harmonizes the acceleration benefits of a structured pattern with the flexibility of fine-grained sparsity [32, 33, 12]. In this study, we focus on N:M semi-structured sparsity within Large Language Models and present a learnable framework to obtain high-quality masks via end-to-end training.

**Learnable Semi-Structured Sparsity.** On another hand, a burgeoning interest exists in developing learnable masks [49, 25, 47], especially in the field of vision models. Markedly contrasted with traditional one-shot pruning methods that rely on a predetermined metric of importance, learnable sparsity can fully leverage the rich information in training data, enabling the identification of more effective sparsity masks. A particularly popular strategy is to directly update the network weight, such as pushing partial weights to zero with Sparse-Refined Straight-Through Estimator (SR-STE) [3, 17, 25] or permuting parameters to achieve better quality [33]. Other methods learn additional indicators to reveal the importance of weight, such as differentiable indexing [35], optimizable combination [47], or decaying [22]. In this work, we make the first attempt to learn N:M masks for frozen LLMs, which is much more challenging due to the huge parameter amount and problem scale.

## 3 Method

### 3.1 N:M Sparsity

We motivate and introduce a learnable framework, MaskLLM, to sparsify Large Language Models (LLMs) for improved inference efficiency. Sparsifying an LLM with N:M patterns imposes the constraint of having (no more than) N non-zero values within each consecutive set of M parameters. This task can be formulated as a mask selection problem with the candidate set of $|\mathbf{S}| = \binom{M}{N} = \frac{M!}{N!(M-N)!}$ candidates, where $|\mathbf{S}|$ denotes the size of the candidate set, and $\binom{M}{N}$ represents the combination number of potential N:M masks. For simplicity, this work primarily focuses on 2:4 sparsity, which can be naturally extended to other patterns such as 1:4 and 4:8. Given a parameter block comprising four consecutive parameters, denoted as $\mathcal{W} \in \mathbb{R}^{1 \times 4}$, the goal of sparsification is to identify the optimal binary mask $\mathcal{M}^* \in \mathbb{B}^{1 \times 4}$ of the same size, ensuring that the pruned weight maintains its behavior on observed data $x \sim p(x)$. For 2:4 sparsity, the binary mask $\mathcal{M}$ must contain exactly two zeros, resulting in a discrete candidate set $\mathbf{S}^{2:4}$ with $|\mathbf{S}^{2:4}| = \binom{4}{2} = 6$ candidates:

$$\mathbf{S}^{2:4} = \{\mathcal{M} \in \mathbb{B}^{1 \times 4} | \sum \mathcal{M} = 2\} = \{\hat{\mathcal{M}}_1, \hat{\mathcal{M}}_2, \hat{\mathcal{M}}_3, \hat{\mathcal{M}}_4, \hat{\mathcal{M}}_5, \hat{\mathcal{M}}_6\} \tag{1}$$

$$= \{[1,1,0,0], [1,0,1,0], [1,0,0,1], [0,1,0,1], [0,1,1,0], [0,0,1,1]\}. \tag{2}$$

For an LLM, there exists a substantial number of parameter blocks, denoted as $\{\mathcal{W}_i\}$, each requiring the selection of corresponding masks $\{\mathcal{M}_i\}$. To maintain satisfactory behavior after pruning, it is natural to define the following objective for N:M sparsity:

$$\{\mathcal{M}_i^*\} = \operatorname*{argmin}_{\{\mathcal{M}_i | \mathcal{M}_i \in \mathbf{S}^{2:4}\}} \mathbb{E}_{x \sim p(x)} \left[ \mathcal{L}_{LM}(x; \{\mathcal{W}_i \odot \mathcal{M}_i\}) \right], \tag{3}$$

where $\mathcal{L}_{LM}$ refers to the language modeling loss for pre-training. The operator $\odot$ denotes element-wise multiplication, which masks partial parameters for sparsification. However, finding the optimal combination of masks $\mathcal{M}^*$ can be extremely challenging in the context of LLMs due to the non-differentiable nature of mask selection and the huge parameter scale. In the following sections, we demonstrate that the mask selection can be transformed into a sampling process.

### 3.2 MaskLLM: Learnable Semi-Structured Sparsity

Consider a single parameter block $\mathcal{W} \in \mathbb{R}^{1 \times 4}$ consisting of only 4 parameters: directly determining the exact optimal mask for this block is not feasible, since the behavior of the pruned LLM also

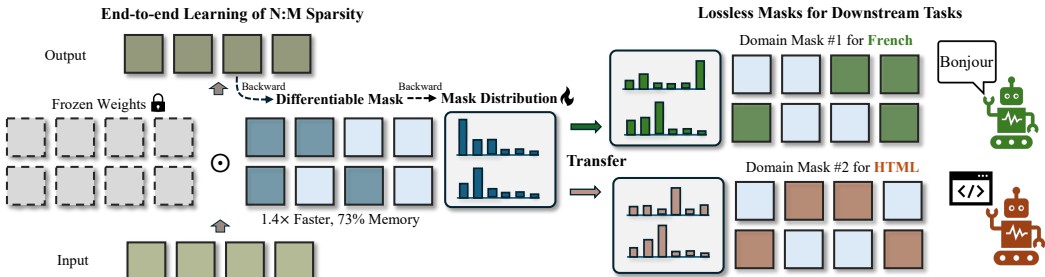

Figure 2: This work introduces learnable semi-structured sparsity for LLMs. MaskLLM models mask selection as a distribution learning problem, enabling the creation of accurate masks through end-to-end training on large-scale datasets. The learned and general mask can be further transferred to downstream tasks or domains, achieving lossless compression.

depends on the pruning of other parameter blocks. Nevertheless, it remains feasible to sample masks independently for each block and assess the overall model quality after pruning. To facilitate random sampling of $\mathcal{M}$, we define a categorical distribution with class probability $p_1, p_2, \ldots p_{|\mathcal{S}|}$, which satisfy $\sum_j p_j = 1$. During the random sampling phase, if a certain mask achieves good quality during pruning, it's reasonable to adjust the categorical distribution by increasing the probability of the sampled mask. With sufficient sampling and updates, this process ends with a distribution where the mask with high probability is more likely to maintain good quality after pruning. Formally, we model the combination problem in Equation 3 from the perspective of random sampling:

$$\{p^*(\mathcal{M}_i)\} = \underset{\{p(\mathcal{M}_i)\}}{\operatorname{argmin}} \mathbb{E}_{x \sim p(x), \mathcal{M}_i \sim p(\mathcal{M}_i)} \left[ \mathcal{L}_{LM}(x; \{\mathcal{W}_i \odot \mathcal{M}_i\}) \right], \quad (4)$$

where $p(\mathcal{M}_i)$ refers to the categorical distribution of $i$-th mask $\mathcal{M}_i$. If it is feasible to get the gradient w.r.t. the distribution, then the above objective can be optimized with gradient descent as demonstrated in Figure 2. Nonetheless, drawing samples from a categorical distribution is still non-differentiable.

**Differentiable Sampling of Masks** An effective method to model a sampling operation is Gumbel Max [14], a re-parameterization trick that disentangles the randomness of sampling into a noise variable. This trick introduces a method to draw samples from the categorical distribution $p$ with an additional noise variable $\epsilon$. It produces the one-hot index $y$ for sampling:

$$y = \operatorname{onehot}(\underset{i}{\operatorname{argmax}}[\log(p_i) + g_i]), \ g_i = -\log(-\log \epsilon_i), \ \epsilon_i \sim U(0, 1), \quad (5)$$

where $\epsilon_i$ is a random noise following uniform distribution, and the $g_i = -\log(-\log \epsilon_i)$ is known as the Gumbel noise. With the Gumbel Max trick, the randomness of sampling is parameterized to an independent variable $g_i$. The only issue towards differentiable sampling lies in the $\operatorname{argmax}$ and onehot operation. To address this, we leverage Gumbel Softmax [21] to approximate the index with Softmax, leading to a soft and differentiable index $\tilde{\mathbf{y}} = [\tilde{y}_1, \tilde{y}_2, \ldots, \tilde{y}_{|\mathbf{S}|}]$:

$$\tilde{y}_i = \frac{\exp((\log(p_i) + g_i)/\tau)}{\sum_j \exp((\log(p_j) + g_j)/\tau)}. \quad (6)$$

The temperature term $\tau$ is a hyper-parameter, controlling the hardness of the sampled index. While $\tau \to 0$, the soft index will be more close to a one-hot vector, resulting in $\tilde{y}_i \to y_i$. With the soft index $\tilde{\mathbf{y}}$ as a row vector and the mask set $\mathbf{S}$ as a matrix where each row $i$ refers to the $i$-th candidate mask $\hat{\mathcal{M}}_i$, it's easy to craft a differentiable mask through a simple matrix multiplication:

$$\tilde{\mathcal{M}} = \tilde{\mathbf{y}} \times \mathbf{S} = \sum_{i=1}^{|\mathbf{S}|} \tilde{y}_i \cdot \hat{\mathcal{M}}_i. \quad (7)$$

This operation produces a weighted average of candidate masks according to the soft index. As shown in Figure 3, we can find all operations, including the sampling and weighted averaging are differentiable, and the gradient w.r.t. the probability $p$ can be easily computed. This allows using the differentiable mask $\tilde{\mathcal{M}}$ to optimize the sampling problem defined in Equation 4.

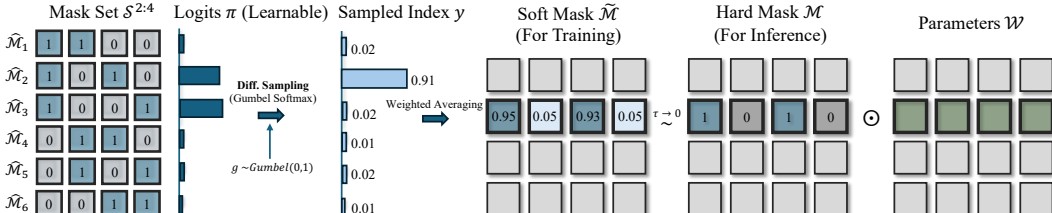

Figure 3: Drawing a random mask from the learnable distribution with Gumbel Softmax. Each consecutive M parameters are associated with a learnable distribution for candidate masks. All illustrated computations, including Gumbel Softmax, and the weighted averaging are differentiable.

**Learning Masks for LLMs**  Equation 7 provides a differentible mask sampled from the underlying distribution $p$. The gradient flow can easily reach the probability $p_i$, making it an optimizable variable in the system. Typically, we do not directly learn the probability and instead, learn the logits $\pi_i$ with a scaling factor $\kappa$, which produces the probability as $p_i = \frac{\exp(\pi_i \cdot \kappa)}{\sum_j \exp(\pi_j \cdot \kappa)}$. As will be discussed in Section 4.3, the scaling factor $\kappa$ will be used to balance the relative magnitude of logits and Gumbel noises, which controls the randomness of sampling. During training, all parameter blocks $\{\mathcal{W}_i\}$ are associated with the corresponding distributions $\{p_\pi(\mathcal{M}_i)\}$, and optimal distribution can be learned in an end-to-end manner. However, our empirical experiments on several large language models reveal a new issue with the learnable masks: the gradient may vanish due to the pruning operation that produces zero parameters in the network. This issue will adversely affect downstream transfer and fine-tuning. To address this, we introduce Sparse Weight Regularization, which maintains an appropriately large magnitude in the remaining weights, leading to the following learning objective:

$$\min_{\{p_\pi(\mathcal{M}_i)\}} \mathbb{E}_{x,\tilde{\mathcal{M}}_i \sim p_\pi(\mathcal{M}_i)} \left[ \mathcal{L}_{LM}(x; \{\mathcal{W}_i \odot \tilde{\mathcal{M}}_i\}) \right] - \lambda \sum_i \|\mathcal{W}_i \odot \tilde{\mathcal{M}}_i\|_2^2. \tag{8}$$

The regularization term weighted by $\lambda$ encourages a large magnitude after pruning.

**Transfer Learning of Sparsity.**  Transfer learning is one of the most popular paradigms in deep learning. In this section, we show the feasibility of transfer learning in sparsity, which crafts new masks by inheriting pre-computed ones. The pre-computed masks can be obtained with oneshot pruning methods like Magnitude Pruning [15], SparseGPT [12] and Wanda [38], or produced by another learning process. Note that given a probability $[p_1, p_2, \ldots, p_{|\mathbf{S}|}]$, the transformation to the final mask is straightforward with a simple $\mathrm{argmax}$. However, if it is possible to map a pre-computed mask back to the class probabilities, then the proposed MaskLLM can begin with a good initialization for sampling. This can hugely improve learning efficiency and quality. To achieve this, we propose Mask Prior, a simple technique to initialize a distribution. Given a prior mask denoted as $\mathcal{M}_0$, we compute its similarity to all candidate masks as:

$$\mathrm{sim}(\mathcal{M}_0, \hat{\mathcal{M}}_i) = \mathcal{M}_0 \hat{\mathcal{M}}_i^\top - \frac{1}{|\mathbf{S}|} \sum_i (\mathcal{M}_i \hat{\mathcal{M}}^\top) = \mathcal{M}_i \hat{\mathcal{M}}^\top - (N/2), \tag{9}$$

which computes the inner product of two masks and re-centers the results with the mean. Note that for N:M sparsity, the range of $\mathcal{M}_0 \hat{\mathcal{M}}_i^\top$ will always be $[0, N]$, the mean value $\sum_i (\mathcal{M}_i \hat{\mathcal{M}}^\top) = N/2$ is a constant. For candidate masks with high similarity to the prior mask, we increase its probability at the initialization stage with:

$$\pi_i' = \pi_i + \sigma(\pi) * \mathrm{sim}(\mathcal{M}_0, \hat{\mathcal{M}}_i) * \alpha, \tag{10}$$

where $\sigma(o)$ is the standard deviation of logits and $\alpha$ is a hyper-parameter that controls the strength of prior. When $\alpha = 0$, we learn the differentiable mask without any prior from one-shot methods.

**Method Summary.**  The learning process of MaskLLM is straightforward. We begin with randomly initialized logits and update it with prior masks as Equation 10 if available. Then we optimize the logits to solve the objective in Equation 8. The mask $\mathcal{M}_i$ with the largest logits will be taken as the final mask for inference. This process is summarized in Algorithm 1.

---

**Algorithm 1** MaskLLM: Learnable Semi-Structured Sparsity for LLMs (2:4)

---

1: **procedure** DIFFERENTIABLEMASK($\pi$, $S$)
2:    Obtain the soft index $\tilde{\mathbf{y}} = \frac{\exp((\pi_i \cdot \kappa + g_i)/\tau)}{\sum_j \exp((\pi_j \cdot \kappa + g_j)/\tau)}$, $g_i = -\log(-\log \epsilon_i)$, $\epsilon_i \sim U(0, 1)$.
3:    Compute the differentiable mask $\tilde{\mathcal{M}} = \tilde{\mathbf{y}} \times \mathbf{S} = \sum_{i=0}^{|\mathbf{S}|} p_i \cdot \mathcal{M}_i$
4:    **return** $\tilde{\mathcal{M}}$
5: **end procedure**
6: $\mathbf{S} = \{\hat{\mathcal{M}}_1, \hat{\mathcal{M}}_2, \ldots, \hat{\mathcal{M}}_{|\mathbf{S}|}\} = \{[1, 1, 0, 0], [1, 0, 1, 0], \ldots [0, 0, 1, 1]\}$
7: ▷ Parallel for all parameter blocks $\mathcal{W}$:
8: Initialize logits $\pi_i \sim \mathcal{N}(0, \sigma)$ for the parameter block $\mathcal{W}$
9: Incorporate prior $\pi_i' = \pi_i + \sigma(\pi) * \text{sim}(\mathcal{M}_0, \hat{\mathcal{M}}_i) * \alpha$ with prior mask $\mathcal{M}_0$
10: **while** Training not terminated **do**
11:    $\tilde{\mathcal{M}} = \text{DifferentiableMask}(\pi, \mathbf{S})$
12:    Update logits $\pi$ with $\nabla_\pi[\mathcal{L}_{LM}(x; \mathcal{W} \odot \tilde{\mathcal{M}}) - \lambda \|\mathcal{W} \odot \tilde{\mathcal{M}}\|_2^2]$
13: **end while**
14: Get the index $k = \text{argmax}(\pi)$
15: Obtain the mask $\mathcal{M}^* = \hat{\mathcal{M}}_k$ for pruning

---

# 4 Experiments

## 4.1 Implementation Details.

We evaluated MaskLLM on three large language model families, ranging in size from 843M to 15B parameters. This included public models like LLaMA-2 7B and 13B [39], Nemotron-4 15B [31], and two in-house models, multilingual GPT-3 843M and 2B [36]. For LLaMA-2 and Nemotron-4, we collected a blended training set following the original papers [36, 31] for training. For the GPT-3 multilingual models, we used the original training set for mask learning. To learn masks, we trained the Gumbel logits for 2,000 steps without updating the LLM parameters. For evaluation, we follow SparseGPT [12] to use C4 dataset [34] for one-shot pruning and Wikitext [28] for evaluation. In addition, we also deploy LM-Eval-Harness [13] for zero-shot evaluation. More details about the models, datasets, training, and evaluation can be found in the appendix.

## 4.2 Learning 2:4 Sparsity in LLMs

> *Finding 1.* Learnable Sparsity scales effectively to large-scale datasets and can fully leverage computational resources to learn precise masks through end-to-end training.

**End-to-end training yields accurate masks.** In Table 1, we report the perplexity and accuracies of our method, compared to three 2:4 sparse baselines: Magnitude Pruning [16], SparseGPT [12], and Wanda [38]. Previous works can produce satisfactory 2:4 masks efficiently but often suffer from inaccurate estimation of weight importance. The inaccuracy mainly arises from two factors: *(1) Accuracy of importance metric*: Due to the difficulty of computing the error caused by pruning, existing methods use approximated metrics to estimate weight importance, which inevitably results in errors. *(2) Scalability*: LLMs are usually pre-trained on large-scale datasets with rich knowledge, but the calibration sets used in existing methods contain very limited samples. With the learnable mask, the above challenges can be naturally addressed through end-to-end training on large-scale datasets, which directly optimizes the language modeling loss. As illustrated in Table 1, MaskLLM yields superior results compared to existing baselines. For instance, with the LLaMA-2 7B model, the proposed method learns a mask with a PPL of 6.72, which is better than the PPL of 10.42 obtained by SparseGPT with weight update. More results such as comparison to other baselines (Table 13) and visualization of mask difference (Figure 8) can be found in the appendix.

**Scaling to large-scale datasets.** To further elaborate on the above analysis, we illustrate the relationship between the number of consumed samples and the Wikitext PPL of pruned LLaMA-2 7B in Figure 4. For one-shot methods such as SparseGPT, all consumed samples are used to compute the Hessian for importance estimation. Increasing the calibration set size from 32 to 256 samples improves the results, but expansion beyond 256 samples yields no notable advantages.

| Method | Wikitext PPL | HellaS. | RACE | PIQA | WinoG. | ARC-E | ARC-C | OBQA | Avg. |
|---|---|---|---|---|---|---|---|---|---|
| **LLaMA-2 7B [39]** | 5.12 | 57.03 | 44.11 | 78.07 | 69.39 | 75.38 | 42.92 | 33.20 | 57.16 |
| - Magnitude | 54.71 | 44.60 | 33.01 | 68.93 | 61.56 | 60.23 | 31.40 | 23.60 | 46.19 |
| - SparseGPT | 10.42 | 43.36 | 36.84 | 71.38 | 63.69 | 62.84 | 29.18 | 22.80 | 47.16 |
| - Wanda | 11.29 | 41.05 | 35.02 | 70.78 | 62.67 | 61.99 | 27.56 | 22.80 | 45.98 |
| - MaskLLM | **6.72** | **50.91** | **40.77** | **74.92** | **64.48** | **69.57** | **36.00** | **28.00** | **52.09** |
| **LLaMA-2 13B [39]** | 4.57 | 60.15 | 44.59 | 79.27 | 72.45 | 78.93 | 47.18 | 34.60 | 59.60 |
| - Magnitude | 8.32 | 48.69 | 38.47 | 70.24 | 59.67 | 61.32 | 29.69 | 22.00 | 47.15 |
| - SparseGPT | 8.20 | 48.62 | 39.62 | 74.54 | **70.00** | 70.29 | 36.00 | 27.20 | 52.32 |
| - Wanda | 8.47 | 46.96 | 38.09 | 74.05 | 66.69 | 68.64 | 34.81 | 25.00 | 50.61 |
| - MaskLLM | **5.85** | **55.09** | **41.24** | **77.69** | 67.80 | **73.15** | **40.44** | **30.00** | **56.74** |
| **Nemotron-4 15B [31]** | 5.78 | 62.60 | 47.75 | 81.34 | 77.11 | 77.69 | 50.77 | 33.00 | 61.47 |
| - Magnitude | 2.78E+03 | 26.30 | 21.91 | 54.62 | 50.67 | 29.29 | 18.52 | 15.60 | 30.98 |
| - SparseGPT | 13.38 | 47.06 | 40.86 | 75.73 | 68.90 | 66.96 | 31.83 | 26.60 | 51.13 |
| - Wanda | 25.05 | 41.13 | 34.16 | 71.71 | 61.72 | 58.46 | 29.78 | 23.80 | 45.82 |
| - MaskLLM | **7.31** | **55.92** | **45.45** | **76.22** | **69.14** | **75.93** | **43.94** | **30.60** | **56.74** |
| **GPT3 2B [36]** | 9.35 | 47.74 | 36.94 | 75.73 | 61.09 | 63.22 | 29.78 | 27.80 | 48.90 |
| - Magnitude | 6.02E+04 | 28.52 | 24.50 | 57.62 | 51.93 | 33.33 | 21.67 | 15.40 | 33.28 |
| - SparseGPT | 22.14 | 34.93 | 29.28 | 66.60 | 54.62 | 53.07 | 22.10 | 16.20 | 39.54 |
| - Wanda | 27.08 | 34.61 | 29.76 | 67.14 | 53.20 | 49.79 | 22.44 | 16.80 | 39.11 |
| - MaskLLM | **11.42** | **42.64** | **33.88** | **73.34** | **58.72** | **57.37** | **26.02** | **21.80** | **44.82** |
| **GPT3 843M [36]** | 12.42 | 39.24 | 33.59 | 70.02 | 54.30 | 53.20 | 21.67 | 21.40 | 41.92 |
| - Magnitude | 1.15E+04 | 25.84 | 21.82 | 55.66 | 50.75 | 28.41 | 19.20 | 15.20 | 30.98 |
| - SparseGPT | 38.78 | 30.26 | 28.33 | 63.38 | 51.62 | 39.27 | 18.86 | 14.80 | 35.22 |
| - Wanda | 51.37 | 30.44 | 28.52 | 61.64 | 49.96 | 40.82 | 18.17 | 14.80 | 34.91 |
| - MaskLLM | **15.39** | **34.65** | **30.91** | **66.76** | **51.86** | **49.07** | **20.48** | **20.00** | **39.10** |

Table 1: Evaluation of 2:4 Sparsity with frozen weights (SparseGPT *does* perform the weight update step). One-shot pruning methods are calibrated with C4 and evaluated on Wikitext-2 following [12]. More results for Llama-3 [1] or other SOTA methods can be found in Table 12 and 13 of the appendix.

In contrast, our proposed learnable method effectively scales to large datasets. Results in Figure 4 show that increasing the number of samples within our framework consistently improves mask quality, with positive results still observable when scaling up to 512k samples. Additionally, our method is also data-efficient and thus applicable to low-resource scenarios with only 1280 samples. With a batch size of 256, the learnable mask is updated for only 5 steps and still produces slightly better masks than SparseGPT. If limited to only one or two steps, the training-based method fails to be comparable to one-shot methods, as this limits the random exploration for finding high-quality masks.

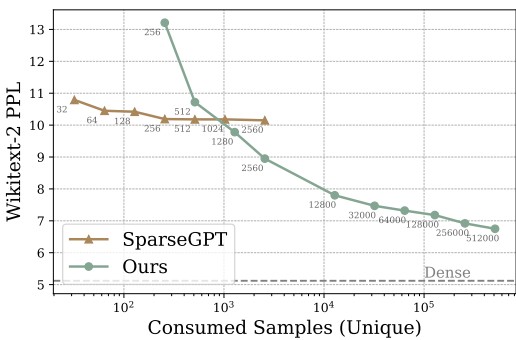

Figure 4: Consumed samples vs. PPL on LLaMA-2 7B. MaskLLM requires 128 samples for the prior and outperforms SparseGPT after 1280 samples.

### 4.3 How to Learn a Good Mask for LLMs

> *Finding 2.* Taking pre-computed masks as prior improves training efficiency and mask quality.

**Transfer Learning with Mask Prior.** An important feature of the proposed method lies in transfer learning. We can initialize the Gumbel logits with pre-computed masks, which significantly accelerate the training. In Table 2, we learn masks using different prior types, including Magnitude prior [15], SparseGPT prior [12], and Wanda prior [38]. Firstly, even without any prior, the learnable mask still achieves superior quality compared to the existing baseline methods, demonstrating its capability to independently discover high-quality masks through end-to-end training. However, learning accurate masks in only 2,000 steps can be challenging due to the massive parameter scale of LLMs. Using prior masks pre-computed by one-shot methods can provide substantial benefits. For example, with the Magnitude prior that can be easily pre-computed according to the weight magnitude, we can improve the wikitext perplexity of LLaMA-2 7B from 9.12 to 6.77.

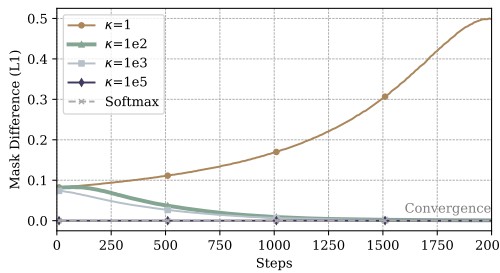
(a) The mask difference between adjacent steps

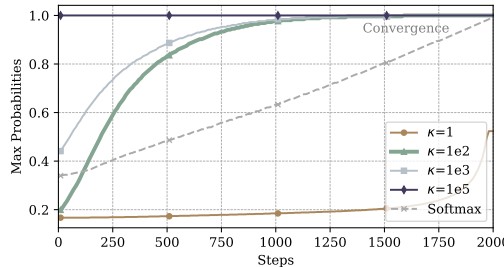
(b) The Maximum probability of mask distribution

Figure 5: (a) The L1 distance of sampled masks between adjacent training steps. (b) The maximum probability of mask distribution, serving as an indicator of convergence. In our method, the randomness of mask sampling is regulated by the scaling factor $\kappa$. A too-small $\kappa$ introduces huge randomness, resulting in slow convergence as shown in (b). And an inappropriately large $\kappa$ will suppress mask exploration and yield zero mask difference throughout the training process in (a).

| Prior Type | GPT-3 843M | | GPT-3 2B | | LLaMA-2 7B | |
|---|---|---|---|---|---|---|
| | Prior Mask | Learned Mask | Prior Mask | Learned Mask | Prior Mask | Learned Mask |
| Magnitude | 1.15E+04 | 16.07 | 6.02E+04 | 12.06 | 54.71 | 6.77 |
| SparseGPT | 79.84 | **15.39** | 24.43 | **11.59** | 10.46 | **6.72** |
| Wanda | 51.37 | 16.39 | 27.08 | 12.18 | 11.29 | 6.80 |
| No Prior | - | 18.62 | - | 14.31 | - | 9.12 |

Table 2: The effectiveness of transfer learning with prior masks. We report the Wikitext PPL of both prior and learned masks. The learned masks use the corresponding prior for initialization and refine the logits through end-to-end training. All results are obtained with frozen weights.

---

> **Finding 3.** The randomness of sampling is crucial for mask learning.

**Encouraging stochastic exploration on candidate masks.** At the early stage of mask learning, the optimal mask is unknown. The stochastic sampling with Gumbel softmax allows for the exploration of different candidate masks, which is crucial for effective learning. As mentioned in Section 3.2, the scaling factor $\kappa$ controls the randomness of sampling. To illustrate this, we visualize the learning process in Figures 5a and 5b, showing the mask difference between adjacent steps and the maximum probability of the learnable distribution, respectively. With a large factor, such as $\kappa$=1e5, the Gumbel softmax will be dominated mainly by the logits rather than the Gumbel noises, which produce similar masks with high confidence throughout the training process. In contrast, with a small scaling factor, such as $\kappa$=1, the Gumbel noises contribute more to the sampling. As illustrated in Figure 5a, the mask is continuously changing during training, leading to slow convergence. Therefore, selecting an appropriate scaling factor is crucial, which should guarantee sufficient randomness and an acceptable convergence speed. In this work, we use a $\kappa$=1e2 and linearly increase it to 5e2 for all experiments.

> **Finding 4.** Maintaining a large magnitude of the remaining weights improves downstream tasks.

**Maintaining a large magnitude of the remaining weights.** In Equation 8, we introduce a regularizer in the form of $-\lambda \sum_i \|\mathcal{W}_i \odot \tilde{\mathcal{M}}_i\|_2^2$. This regularizer is crucial for both mask learning and transfer learning, as it directly influences the magnitude of gradients during training. For instance, if certain layers are pruned to a small magnitude, the gradients passed to their inputs will also diminish, thereby impeding mask learning and transfer to downstream tasks. In Table 3, we demonstrate the effectiveness of weight regularization under different scenarios, such as mask training, LLM fine-tuning after pruning, and transfer learning to downstream tasks. As will be elaborated in subsequent sections, the proposed regularization helps the learning of lossless masks for downstream tasks. We provide more analysis in Section F of the Appendix.

| Task (2B) | w/o Reg. | w/ Reg. |
|---|---|---|
| Mask-only | 11.59 | **11.42** |
| Sub-domain | 7.61 | **7.39** |
| Finetuning | 10.21 | **9.96** |

Table 3: Weight Regularization on remaining weights helps mask learning

| Domain | C# | HTML | Pascal | Story | French | Japanese | Chinese | OpenWeb | Average |
|---|---|---|---|---|---|---|---|---|---|
| **GPT3-2B Dense** | 1.78 | 1.54 | 2.50 | 14.76 | 9.71 | 8.75 | 8.25 | 12.05 | 7.42 |
| - Magnitude | 1.38E+03 | 1.72E+03 | 1.64E+03 | 2.12E+05 | 6.950E+02 | 5.67E+02 | 7.22E+02 | 1.23E+05 | 4.27E+04 |
| - SparseGPT | 2.54 | 2.41 | 3.86 | 30.37 | 26.99 | 28.69 | 26.93 | 28.66 | 18.80 |
| - SparseGPT-Update | 2.20 | 2.11 | 3.09 | 25.43 | 20.35 | 22.34 | 20.55 | 26.69 | 15.36 |
| - Wanda | 2.86 | 2.68 | 4.75 | 40.07 | 31.37 | 36.75 | 33.03 | 35.34 | 23.36 |
| - MaskLLM | **1.76** | **1.54** | **1.94** | **15.58** | **9.61** | **7.96** | **6.92** | **13.84** | **7.39** |

| Domain | CUDA | VHDL | Javascript | BigScience | Reddit-Plus | Book | Arxiv | MedAbs | Average |
|---|---|---|---|---|---|---|---|---|---|
| **LLaMA2-7B Dense** | 1.74 | 1.86 | 2.01 | 6.28 | 11.05 | 7.02 | 3.49 | 4.95 | 4.80 |
| - Magnitude | 9.92 | 13.60 | 2.52 | 66.80 | 81.56 | 72.95 | 32.17 | 29.31 | 38.60 |
| - SparseGPT | 2.09 | 2.30 | 2.52 | 9.57 | 15.46 | 9.91 | 4.54 | 6.73 | 6.64 |
| - SparseGPT-Update | 1.91 | 2.08 | 2.32 | 9.63 | 14.52 | 9.78 | 4.21 | 6.14 | 6.32 |
| - Wanda | 2.32 | 2.59 | 2.80 | 11.56 | 18.62 | 12.83 | 5.23 | 8.36 | 8.04 |
| - **MaskLLM** | **1.80** | **1.83** | **2.01** | **6.88** | **10.12** | **8.10** | **3.51** | **4.95** | **4.90** |

Table 4: Learning customized masks for downstream tasks with frozen LLM weights.

| Mask Type (2B) | Avg. Task PPL |
|---|---|
| Dense | 7.42 |
| General Mask | 10.61 |
| Scratch Mask | 7.51 |
| Transfer Mask | **7.39** |

Table 5: Transfer learning is effective for downstream tasks.

| Methods | Storage per Task (bits per param) | Model Size in Memory | Speed |
|---|---|---|---|
| Finetuning | 16 | 100% | 1.0× |
| Learned 2:4 masks | **0.65** (↓ 25×) | 73% | **1.4×** |

Table 6: Storage and inference cost of of llama-2 7B for downstream tasks

## 4.4 Learning N:M Sparsity for Downstream Tasks

> ***Finding 5.*** Learned masks can losslessly adapt frozen LLMs to downstream tasks, offering a 1.4× wall clock GPU speed up and 73% memory footprint.

Large language models can achieve satisfactory quality across a variety of tasks. In many cases, we are more interested in one particular ability of these large models under a specific task, such as programming or translation, for which an LLM is over-parameterized. This naturally introduces a new problem: can we learn a mask for specific tasks to achieve lossless compression? To evaluate this, we learn masks for 2,000 steps separately on different domains and tasks and report the task-wise PPL in Table 4. We considered one-shot pruning as baselines, where we collected 256 samples from the task dataset for calibration. Results show that lossless masks can be learned for many tasks with our method.

We also evaluated the power of transfer learning for downstream tasks in Table 5. To deploy sparse LLMs for a single task, we can directly pick the pre-computed general mask from Table 1, or train an "expert" mask from scratch. However, both strategies show a quality drop compared to the dense model (PPL=7.42) since they either allocate some capacity for other domains (PPL=10.61) or only see limited data from target domains (PPL=7.51). Our work leverages the general mask as prior and transfers it to the downstream tasks, which can produce lossless models (PPL=7.39).

Updating parameters for downstream tasks results in additional copies of the model for each task, incurring higher storage costs. Learning masks alone allows for encoding task-specific masks with minimal space while keeping only a single, shared copy of the original parameters. As shown in Table 6, task-specific masks only need 0.65 bits per parameter for storage on disk using simple arithmetic coding[3] with a static, uniform symbol distribution. For BS=1 inference on an A6000 GPU, 2:4 sparsity brings 1.4× acceleration and 27% reduction in the memory footprint (broader speedup results appear in Table 16 in the appendix).

## 5 Conclusion

In this work, we present MaskLLM, a learnable pruning method that crafts accurate N:M sparsity in LLMs, thereby reducing computational overhead during inference. Our empirical experiments on several models show the scalability of MaskLLM to large-scale data and the effectiveness of end-to-end training for mask learning. Furthermore, we demonstrate that lossless compression with N:M sparsity is attainable in downstream tasks, underscoring its practicality for real-world applications.

---

[3]https://pypi.org/project/arithmetic-compressor/     ($\log_2(6)/4$ bits for 6 mask candidates per group)

## Acknowledgments and Disclosure of Funding

This work is in part supported by the Singapore Ministry of Education Academic Research Fund Tier 1 (WBS: A-0009440-01-00). We would like to thank Jorge Albericio Latorre for the fruitful discussion and feedback on the project.

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

# A    Implementation Details

Here we provide more details about the models, training data, training configurations and other resources used in our experiments.

**LLaMA-2**  For LLaMA-2, we collected a blended training set following the official paper [39], which consists of corpuses from 69 domains, covering CUDA, VHDL, Reddit etc. For training, we selected a subset of 512k samples from the dataset and updated the learnable mask for 2,000 steps, with a global batch size of 256. We used 64 A100 GPUs during training with an 8-way tensor parallel configuration. The full training took 1,280 GPU hours for LLaMA-2 7B and 2,304 GPU hours for LLaMA-2 13B. In Table 11, we also provide training results solely using the C4 dataset.

**Nemotron-4**  For Nemotron-4, we collected a small training dataset covering three domains: CC-MAIN-2021-31, Open Web Math, and Gutenberg Fuzzy. We used a subset of 512k samples and trained the model with 64 A100 GPUs using an 8-way tensor parallel configuration. The training process took 2,304 GPU hours.

**GPT-3 (An Internal LLM).**  The GPT-3 multilingual models were pre-trained using the Megatron framework on a corpus of 1.1 trillion tokens. These models share a similar network architecture with the official GPT [5], utilizing the standard transformer architecture [40] with layer normalization, SwiGLU activation function, and Rotary Positional Embeddings (ROPE) [37]. Both the 2B and 843M parameter models comprise 24 transformer layers with 16 attention heads. The hidden sizes are 2048 for the 2B model and 1024 for the 843M model. Furthermore, the maximum sequence length for these models is 4096 tokens. For pre-training, a multilingual dataset was collected, encompassing 110 domains such as HTML, C++, French, etc.

# B    Hyper-parameters

We summarize the hyper-parameters used in our experiments in Table 7. The main results of hyper-parameter tuning are available in Table 10, where we assessed different temperature, logit scaling factors and prior strength with GPT-3 843M.

| Model | Optimizer | Training Steps | Logits Init | Scaling Factor $\kappa$ | Gumbel Temp. $\tau$ | Prior and Strength $\alpha$ | Sparse Reg. |
|---|---|---|---|---|---|---|---|
| LLaMA-2 7B | AdamW(5e-4, wd=0.1) | 2,000 | $\mathcal{N}(0, 0.01)$ | [1e2, 5e2] | [4, 0.05] | SparseGPT($\alpha$=3) | 1e-5 |
| LLaMA-2 13B | AdamW(5e-4, wd=0.1) | 2,000 | $\mathcal{N}(0, 0.01)$ | [1e2, 5e2] | [4, 0.05] | SparseGPT($\alpha$=3) | 1e-5 |
| LLaMA-3 8B | AdamW(5e-4, wd=0.1) | 2,000 | $\mathcal{N}(0, 0.01)$ | [1e2, 5e2] | [4, 0.05] | SparseGPT($\alpha$=3) | 1e-5 |
| Nemotron-4 14B | AdamW(5e-4, wd=0.1) | 2,000 | $\mathcal{N}(0, 0.01)$ | [1e2, 5e2] | [4, 0.05] | SparseGPT($\alpha$=3) | 1e-5 |
| GPT-3 2B | AdamW(1e-3, wd=0.1) | 2,000 | $\mathcal{N}(0, 0.01)$ | [1e2, 5e2] | [4, 0.05] | SparseGPT($\alpha$=3) | 1e-5 |
| GPT-3 843M | AdamW(1e-3, wd=0.1) | 2,000 | $\mathcal{N}(0, 0.01)$ | [1e2, 5e2] | [4, 0.05] | SparseGPT($\alpha$=3) | 1e-5 |

Table 7: Training details and hyper-parameters for mask training

| Temperature $\tau$ | $1 \to 0.05$ | $2 \to 0.05$ | $4 \to 0.05$ | $10 \to 0.05$ |
|---|---|---|---|---|
| **Wikitext PPL** | 17.52 | 16.69 | 15.39 | 15.68 |

Table 8: Hyper-parameter tuning on GPT-3 843M for Gumbel softmax temperature

| Scaling Factor $\kappa$ | $1 \to 5$ | $1e2 \to 5e2$ | $1e3 \to 5e3$ | $1e5 \to 5e5$ |
|---|---|---|---|---|
| **Wikitext PPL** | 5.97E+06 | **15.39** | 15.59 | 24.81 |

Table 9: Hyper-parameter tuning on GPT-3 843M for the logit scaling factor

| Prior Strength $\alpha$ | 0 | 1 | 2 | 3 | 5 |
|---|---|---|---|---|---|
| **Wikitext PPL** | 18.62 | 23.65 | 15.59 | **15.39** | 15.48 |

Table 10: Hyper-parameter tuning on GPT-3 843M for the logit prior strength

## C   Mask Learning with the C4 Dataset

In table 11, we compare the learned masks on the C4 dataset [34] with those on the blended datasets discussed in Section A. Our blended dataset encompasses a broader range of topics and domains compared to the C4 dataset, including coding, different languages, etc. Despite this, the result in table 11 still indicates that MaskLLM is able to learn accurate masks on the C4 dataset, with a minor difference ($\Delta$PPL=0.07) compared to the result obtained on the blended dataset.

| Method | Blended Data (See Sec. A) | C4 [34] |
|---|---|---|
| Llama-2 7B | 5.12 | 5.12 |
| SparseGPT [12] | 9.88 | 10.42 |
| Wanda [38] | 11.25 | 10.29 |
| MaskLLM | **6.72** | **6.79** |

Table 11: Wikitext-2 PPL of 2:4 LLaMA-2 7B pruned with different datasets

## D   2:4 Results on Llama-3 8B

In table 12, we present additional pruning results for Llama-3 8B [1], adhering to the same training protocol as described in Table 7. For reproducibility, we utilize the C4 dataset for both calibration and mask learning.

| Method | Weight Update | Wikitext-2 PPL |
|---|---|---|
| Llama-3 8B Dense | - | 5.76 |
| Magnitude [17] | - | 2.61E+03 |
| SparseGPT [12] | ✓ | 17.64 |
| Wanda [38] | - | 23.40 |
| MaskLLM | - | **8.50** |

Table 12: Wikitext-2 PPL of 2:4 LLaMA-3 8B, with the sequence length of 4096. We took the SparseGPT mask as the prior and learned the mask on the C4 dataset.

## E   Comparison to More Pruning Methods for LLMs

In Table 13, we compare MaskLLM to several baseline methods that were not implemented using the Megatron framework. We report the official results on Wikitext-2 PPL and LLaMA-2 13B. Even compared to methods that incorporate weight updates, our method achieves superior perplexity results.

| Method | Weight Update | Wikitext-2 PPL |
|---|---|---|
| Llama-2 13B Dense | - | 4.57 |
| SparseGPT [12] | ✓ | 8.32 |
| Wanda [38] | - | 8.27 |
| ADMM-Iter [4] | ✓ | 7.78 |
| GBLM [8] | - | 8.80 |
| RIA [46] | - | 8.41 |
| Pruner-Zero [9] | - | 7.41 |
| MaskLLM | - | **5.85** |

Table 13: Comparison to SOTA 2:4 pruning methods on LLaMA-2 13B, with all results collected from original papers or official implementations.

# F Sparse Weight Regularization

**Weight Norm of Sparse LLMs.** In Equation 8, we introduce an additional term to preserve sufficient gradients during training. As shown in Table 14, a larger weight regularization facilitates large gradients during training, which is beneficial for mask exploration. We also illustrate the weight magnitude of pruned LLMs, obtained using magnitude pruning, SparseGPT (Hessian) and MaskLLM in Figures 6a and 6b. These figures show the relative L1 norm of pruned weights compared to the magnitude pruning baseline, which produces the largest weight norm after pruning. An interesting observation is that, even when initialized with a magnitude prior, learnable method may still select some smaller values during pruning, resulting in a 10% lower norm than magnitude pruning. Introducing sparse weight regularization can effectively improve the weight norm of sparse LLMs and enhance their quality in further transfer learning or fine-tuning for downstream tasks.

| Regularization | Average Gradient Norm |
|:---:|:---:|
| 0 | 0.219 |
| 1e-5 | 0.542 |
| 1e-4 | 0.559 |

Table 14: Average Gradient Norm over the First 500 Training Steps of GPT-3 2B, with Varying Levels of Sparse Weight Regularization. In this study, we use a regularization strength of 1e-5 for mask learning, as it offers a stable gradient while imposing minimal constraints on the search space.

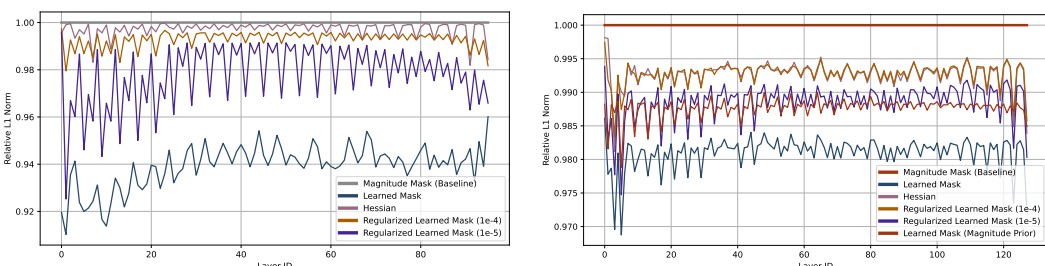

(a) Relative norm of remaining weights (GPT-3 2B).     (b) Relative norm of remaining weights (LLaMA2 7B).

Figure 6: The relative L1 norm of pruned weights compared to magnitude pruning

# G Layer Sensitivity

**Sensitivity Analysis with Learnable Method.** In Figure 7, we analyze the sensitivity of LLaMA-2 7B using both the learnable and one-shot methods. For efficiency, we update the learnable masks for 500 steps and use Wikitext PPL as the metric. We observe a similar trend in the learned masks and SparseGPT masks, suggesting that a fast one-shot pruning method can reliably indicate sensitivity. Additionally, for 2:4 sparsity, the last layer is typically more sensitive than other layers. To maintain satisfactory results, we can keep the last layer dense, achieving a good trade-off between efficiency and quality. In Table 15, we report the pruning results when a few layers are kept dense.

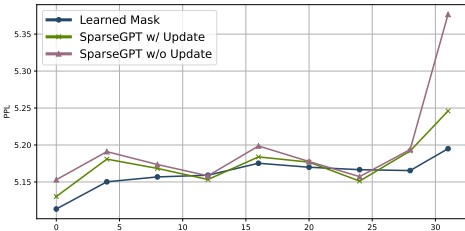

Figure 7: Layer Sensitivity of LLaMA-2 7B

| Strategy | Story | OpenWeb |
|:---|:---:|:---:|
| Dense | 14.76 | 12.05 |
| Full Sparsity | 15.58 | 13.84 |
| Skip the last 1 layer | 15.18 | 13.61 |
| Skip the first 1 layer | 15.41 | 13.88 |
| Skip the last 4 layers | 15.07 | 13.24 |
| Skip the last 8 layers | 14.95 | 12.92 |

Table 15: Keeping sensitive layers dense can be a way to trade-off quality and efficiency.

## H    Throughput of 2:4 LLaMA-2 7B.

In Table 16, we benchmark the throughput of LLaMA-2 7B with 2:4 sparsity on an A6000 GPU using TensorRT-LLM for a batch size of 1. Throughput is evaluated as the number of tokens processed per second. Over a variety of input and output lengths, 2:4 sparsity achieves an overall acceleration of $1.36\times$ to $1.41\times$ compared to the dense model.

| Model | Input Len. | Output Len. | Throughput (Token/s/GPU) | | |
|---|---|---|---|---|---|
| | | | Dense | 2:4 | Speed Up |
| Llama-2 7B | 128 | 128 | 61.74 | 86.92 | $1.41\times$ |
| | 128 | 2048 | 59.18 | 82.11 | $1.39\times$ |
| | 2048 | 128 | 57.55 | 78.81 | $1.37\times$ |
| | 2048 | 2048 | 55.40 | 75.10 | $1.36\times$ |
| Llama-2 13B | 128 | 128 | 32.97 | 51.64 | $1.57\times$ |
| | 128 | 2048 | 31.81 | 49.00 | $1.54\times$ |
| | 2048 | 128 | 31.14 | 47.19 | $1.55\times$ |
| | 2048 | 2048 | 30.09 | 45.06 | $1.50\times$ |

Table 16: Benchmarking LLaMA-2 7B and 13B on A6000 with TensorRT-LLM.

## I    Mask Difference

In Figure 8, we visualize the differences between learned masks and one-shot masks, using SparseGPT as the prior for the learnable mask. We observe that SparseGPT and Wanda produce similar masks, with differences typically ranging from 5% to 10%, due to their similar pruning objectives. Our method, however, can produce distinct masks compared to these baselines, as shown in Figures 8a and 8b. Additionally, we find that weight regularization is crucial for effective mask learning. Without weight regularization, the vanished gradient can hinder mask learning, resulting in only a 2.83% difference from the prior, as shown in Figure 8c.

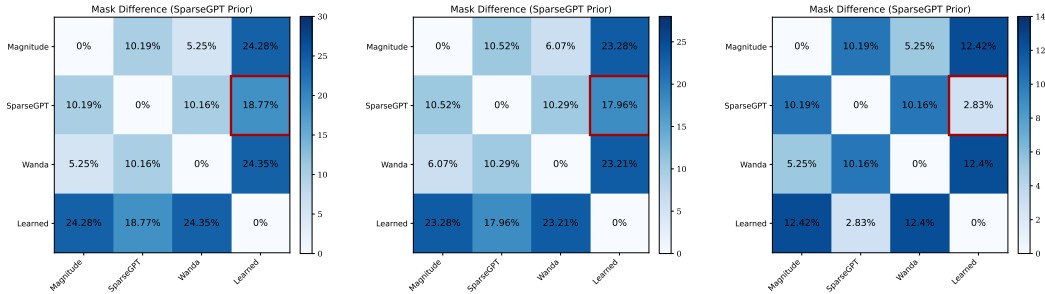

(a) Mask Difference on LLaMA-2 7B with regularization

(b) Mask Difference on GPT-3 2B with regularization

(c) Mask difference on LLaMA-2 7B without regularization.

Figure 8: (a) & (b) Notable mask difference exists between learned masks and one-shot ones. (c) without sparse weight regularization, the vanished gradient caused by pruning will hinder mask training, leading to a low difference between prior masks and learned masks.

## J    MaskLLM for Vision Transformers

To evaluate the generalizability of learnable semi-structured sparsity, we further extend the proposed method to Vision Transformers (ViTs) [10]. Table 17 presents the top-1 accuracy on ImageNet-1K by pruning an off-the-shelf ViT-B/16. In the one-shot pruning scenario, we randomly selected 128 samples from the training dataset as the calibration set, and no additional fine-tuning was conducted post-pruning. For MaskLLM, we utilized the SparseGPT mask as a prior and directly optimized the learnable mask on the ImageNet dataset while keeping the model weights frozen. Remarkably, with just a single epoch of optimization, the learnable mask achieved an accuracy of 76.23%, significantly outperforming SparseGPT's 71.52%. Moreover, a fully optimized mask with 20 epochs of training

| Method | Sparsity Pattern | Weight Update | Top-1 Acc. (%) |
|---|---|---|---|
| ViT-B/16 | Dense | - | 79.15 |
| Magnitude | 2:4 | - | 65.92 |
| Wanda | 2:4 | - | 63.28 |
| SparseGPT | 2:4 | ✓ | 71.52 |
| SparseGPT w/o Update | 2:4 | - | 59.72 |
| MaskLLM-4V (1 Epoch) | 2:4 | - | 76.23 |
| MaskLLM-4V (20 Epochs) | 2:4 | - | **79.46** |

Table 17: MaskLLM for Vision Transformers

achieved lossless compression ($\Delta$Acc = +0.31%) for the ViT-B/16 model. This observation also suggests the presence of a Lottery Ticket phenomenon [11] in modern Vision Transformers, where a sparse sub-network can match the original model's performance without any weight update.

# K  Limitations.

In this work, we explore an end-to-end learning method for semi-structured pruning. Although our method yielded superior results, training LLMs with learnable masks inevitably consumes more resources compared to one-shot methods, which can produce masks efficiently. Improving the training efficiency of learnable masks is an important topic in future works.

# L  Broader Impacts.

The technique proposed in this paper will not lead to negative societal impact. On the contrary, it offers significant benefits, including the reduction of energy costs and carbon emissions associated with the deployment of Large Language Models. By optimizing for Semi-structured (or 'N:M') Sparsity, our method reduces the computational resources required for inference, thereby contributing to more sustainable and environmentally friendly AI applications.

