# OpenReview forum: "MaskLLM: Learnable Semi-Structured Sparsity for Large Language Models"
_NeurIPS.cc/2024/Conference — NeurIPS 2024 spotlight_

### Official Review · Reviewer_qjsy · 2024-06-16

**Soundness:** 4
**Presentation:** 4
**Contribution:** 4
**Rating:** 8
**Confidence:** 5

**Summary:**

This paper introduces the concept of learnable semi-structured sparsity (N:M sparsity). This extends to practical N:M sparsity to pretrained LLMs and makes the masks learnable. the paper also make the transfer of masks from other sparsity techniques to work for N:M. Finally, the general N:M masks learned can be easily fine-tuned for each task with great performance. the paper also shows extensive evaluations on various practical LLMs with the SOTA performance compared to other sparse LLM methods.


----------------------
The review will be short and does not reflect the time put in for the review or the quality of the paper. When the ideas are simple and clear -- I tend to write shorter reviews to the point.

**Strengths:**

I really enjoyed reading this paper. It was a very practical paper on many levels I have read on sparsity and applicability to LLMs.

The idea of differentiable masks is not new -- as pointed by the authors -- in CS (Savarese et al., 2019), STR (Kusupati et al., 2020) along with other methods mentioned in the paper. Same goes with N:M. However, bringing them together (again done earlier as mentioned by the paper) and combining them with pretrained LLMs makes is very useful in practice. I really like 2 other aspects apart from the base idea and practicality. THe transfer of one-shot pruning masks to N:M as scaffolds and then using learned N:M masks for downstream tasks was strong.

The results are pretty solid as well. I also like the downstream evals and not just having perplexity values. the findings of the paper highlighted are very useful in further research.

I am strongly in support of the paper unless I missed something obvious. I appreciate the author on the comprehensive paper.

**Weaknesses:**

I do not find any glaring weaknesses concerns or questions about the current version of the paper. However, I might have missed something and would rely on the other reviewers if I was wrong.

My only suggestion for the paper is to add a more comprehensive related work section to complete the paper. Maybe introducing learning sparsity be it mask learning with STE or STR would be a nice thing to add.

**Questions:**

See above.

**Limitations:**

Yes

---

> ### Author Rebuttal · Authors · 2024-08-06
>
> Thank you so much for the invaluable suggestions and the positive comments! We will polish our draft with the following new results and make our code and learned masks public for better reproducibility.
>
> > **Q1: My only suggestion for the paper is to add a more comprehensive related work section to complete the paper. Maybe introducing learning sparsity be it mask learning with STE or STR would be a nice thing to add.**
>
> **A1:** Thanks for the suggestion. We provide new results in the following table by re-implementing the SR-STE method [1]. Additionally, we compare MaskLLM to several fine-tuning methods [2,3] from existing works, including simple fine-tuning and PEFT.
> This table indicates that MaskLLM, even without any weight updates, can achieve competitive results (PPL=6.72) compared to SR-STE (PPL=6.80). This result is expected, as MaskLLM can explore all mask candidates, whereas SR-STE primarily focuses on weights with large magnitudes. Additionally, incorporating sparse fine-tuning into MaskLLM can significantly improve both perplexity (PPL) and zero-shot accuracy on HellaSwag.
>
> We will polish the related work section and the appendix to include those baselines, and will also make our code and learned masks public for better reproducibility.
>
> |Method|Weight Update|LLM|Dense PPL↓|Sparse PPL↓|Δ PPL ↓|HellaSwag ↑|
> |:-|:-:|:-:|:-:|:-:|:-:|:-:|
> |Wanda + Sparse Finetuning [1]|✔️|Llama-1 7B|5.68|7.02|+1.33|n/a|
> |Wanda + LoRA [1]|✔️|Llama-1 7B|5.68|8.24|+2.56|n/a|
> |SR-STE [2]|✔️|Llama-2 7B|5.12|6.80|+1.68|51.26|
> |SparseGPT + SPP [3]|✔️|Llama-2 7B|5.12|n/a|n/a|51.33|
> |Wanda + SPP [3]|✔️|Llama-2 7B|5.12|n/a|n/a|50.61|
> |**MaskLLM + Sparse Finetuning**|✔️|**Llama-2 7B**|**5.12**|**5.83**|**+0.71**|**54.66**|
> |**MaskLLM**|-|**Llama-2 7B**|**5.12**|**6.72**|**+1.60**|**50.91**|
>
> *Table 1: Comparison to more finetuning-based baselines.*
>
>
> [1] Learning N:M Fine-grained Structured Sparse Neural Networks From Scratch
> [2] A Simple and Effective Pruning Approach for Large Language Models
> [3] SPP: Sparsity-Preserved Parameter-Efficient Fine-Tuning for Large Language Models

---

> > ### Comment · Reviewer_qjsy · 2024-08-07
> >
> > Thanks for the rebuttal. After looking at other reviews as well, I am in support of accepting this paper.
> >
> >
> > Please ensure to update the related work section and incorporate the new experiments presented here in the final paper.

---

> > > ### Author Response · Authors · 2024-08-10
> > >
> > > Thank you for the very positive comments and invaluable suggestions! We will incorporate all the mentioned results into the draft as suggested.
> > >
> > > Best regards,
> > > Authors of #1520

---

### Official Review · Reviewer_HusL · 2024-07-11

**Soundness:** 3
**Presentation:** 3
**Contribution:** 3
**Rating:** 8
**Confidence:** 4

**Summary:**

This paper proposes MaskLLM, a learnable method to craft semi-structured sparsity in Large Language Models (LLMs). The approach involves modeling N:M masks with a categorical distribution, which is can be optimized through gumbel softmax. The key findings in this paper suggest that end-to-end learning of mask parameters can learn more effective and accurate sparsity patterns compared to one-shot methods. Additionally, the porposed MaskLLM also supports transfer learning for downstream tasks, where lossless masks can be learned for deployment.

**Strengths:**

1. The ideas of learnable masks and transfer learning are innovative for Sparse LLMs. The proposed method enables task-oriented compression for downstream applications without necessitating the re-training of LLM weights, making it practical for real-world applications.
2. Results on several LLMs are positive. Table 1 demonstrates that the learnable mask method can achieve superior performance compared to state-of-the-art methods, while keeping the LLM weights frozen throughout the learning process. This indicates significant potential for further improvements in N:M sparsity within LLMs.
3. The results in Table 4 are interesting. They show that the learnable mask does not require so many training steps and samples to outperform the oneshot baseline. And the proposed method is more scalable with large-scale datasets.
4. The key ideas and findings in this paper are clear and well-organized. It is easy to understand the key messages in different experiments.

**Weaknesses:**

1. One of my concerns is the use of non-public datasets and LLMs in this study. It would be beneficial for the paper to include more results using publicly available data and open-source models, such as LLaMA-3, to enhance the reproducibility and applicability of the findings.
2. This is a question about the storage cost in Table 6: Could the author clarify the actual storage costs, such as the file size on disk? Additionally, what is the data format used during the fine-tuning process?
3. Table 2 highlights the significance of the prior in mask selection. Does this suggest that the proposed method is only trying to fine-tune the prior mask rather than learn something new? What happens if we don’t have SparseGPT prior? Is the learned mask very similar to its prior?

**Questions:**

Please refer to "Weaknesses".

**Limitations:**

Limitations are adequately addressed.

---

> ### Author Rebuttal · Authors · 2024-08-06
>
> Thanks for the invaluable comments and questions!
>
> > **Q1: One of my concerns is the use of non-public datasets and LLMs in this study. It would be beneficial for the paper to include more results using publicly available data and open-source models, such as LLaMA-3, to enhance the reproducibility and applicability of the findings.**
>
> **A1:** Thank you for the suggestions. We provide additional results for the public C4 dataset and Llama-3 7B.
>
> **1) The C4 dataset:** The following table demonstrates that our method achieves comparable results across different datasets. However, the blended dataset yielded slightly better results due to its inclusion of more diverse domains, such as programming.
>
> |Method|Blended Data|C4|
> |-|:-:|:-:|
> |Llama-2 7B|5.12|5.12|
> |SparseGPT|9.88|10.42|
> |Wanda|11.25|10.29|
> |MaskLLM|**6.72**|**6.85**|
>
> *Table 1: Wikitext-2 PPL of 2:4 LLaMA-2 7B pruned with different datasets.*
>
>
> **2) Llama-3 8B**: In addition, we also provide new results for Llama-3 8B only using the public C4 dataset, using the same training protocol and hyperparameters as Llama-2 7B for mask learning. Our method continues to achieve superior performance on the latest Llama-3 model. We will include all these new results in the appendix.
> | Method           | Weight Update | Wikitext-2 PPL |
> |----|:-:|:-:|
> | Llama-3 8B Dense | -             | 5.76           |
> | Magnitude        | -             | 2.61E+03       |
> | SparseGPT        | ✔️             | 17.64          |
> | Wanda            | -             | 23.40          |
> | MaskLLM          | -             | **8.50**       |
>
> *Table 2: Wikitext-2 PPL of 2:4 LLaMA-3 8B, with a sequence length of 4096. We took the SparseGPT mask as the prior and only used the public C4 dataset for this experiment.*
>
> > **Q2: This is a question about the storage cost in Table 6: Could the author clarify the actual storage costs, such as the file size on disk? Additionally, what is the data format used during the fine-tuning process?**
>
> **A2:** Regarding the Llama-3 model, the actual storage cost for the binary mask is 564MB, which is compressed using the lossless compression method -- ``np.savez_compressed``. In contrast, the original model requires 15GB for storage. For five downstream tasks, using SparseGPT would require 5x (15GB/2 (for 50% zeros) + 564MB) on disk, but MaskLLM would require only 15GB + 5x 564MB. The training is conducted with BF16 precision.
>
> > **Q3: Table 2 highlights the significance of the prior in mask selection. Does this suggest that the proposed method is only trying to fine-tune the prior mask rather than learn something new? What happens if we don’t have SparseGPT prior? Is the learned mask very similar to its prior?**
>
> **A3:** Thank you for the question. A mask prior can be a useful jump-start, helping to reduce the number of samples needed to achieve good quality. MaskLLM indeed inherits some masks from the prior and refines them to enhance performance. Figure 8 in the appendix illustrates the mask differences. A small difference is observed between magnitude pruning and other one-shot methods like SparseGPT and Wanda. However, after training, the mask difference between MaskLLM and one-shot methods can be larger than the differences among the one-shot methods themselves. Regarding the availability of different priors, we believe that the magnitude prior is always accessible for mask learning, showing comparable or even superior performance to other priors like SparseGPT or Wanda.

---

> > ### Comment · Reviewer_HusL · 2024-08-11
> > **Response to Rebuttal**
> >
> > I have read through the response from the authors. The authors provided sufficient experiment results and statistics in the rebuttal to support their claim. My three concerns are adequately addressed. I have also read through other reviews that proves the work is rather solid. I will raise my score.

---

> > > ### Author Response · Authors · 2024-08-11
> > >
> > > Thanks so much for the encouraging feedback! We will keep refining the draft with all the additional experiments.
> > >
> > > Best regards,
> > > Authors

---

### Official Review · Reviewer_2tsd · 2024-07-11

**Soundness:** 3
**Presentation:** 2
**Contribution:** 3
**Rating:** 7
**Confidence:** 4

**Summary:**

The paper introduces MaskLLM, a method for introducing semi-structured sparsity (N:M mask patterns) in LLMs. The authors show that existing model pruning methods such as SparseGPT result in significant loss in model quality at smaller scales (800M ~ 15B parameters) when using semi-structured methods. They formulate the problem of find a good N:M pattern as a mask selection problem from a candidate set of mask $S$. They formulate this further as a sampling problem, whereby they sample masks for all $\mathcal{W}^{1\times4}$ parameters in a layer to measure model quality - since exact mask computation is an intractable problem for large models. They propose using Gumbel-softmax sampling to figure out soft / differentiable masks that can learned through a training process, to optimize the sampling problem. The authors further find that learning of the masks by their proposed method can result in vanishing gradients throughout the network, and enable a sparse weight regularization to promote higher gradient norms through the network. They further find that using methods such as SparseGPT as mask priors enables more effective sampling (ie, learning) of the final masks. They follow these up with different experiments and ablations to highlight the strengths of their proposed method.

**Strengths:**

1. The paper formulates the selection of N:M masks as an optimization problem, which enables scaling the method to large datasets and models.
2. The paper walks through the math required to understand their optimization formulation step-by-step, making it simple to understand.
3. The method focuses on common problems introduced by pruning (such as vanishing gradients of pruned weights etc.) and proposes ways to resolve them through regularization.
4. For each of the parts of the proposed method, the authors present ablations and results that validate their design choices.
5. The authors also showcase how their method can scale for transfer learning to downstream tasks (both via fine-tuning or via mask-transfer, as proposed via their prior initialized method)
6. The paper shows performance results from using their method (~1.3~1.4x faster) on A6000 GPUs.

**Weaknesses:**

1. The best MaskLLM results presented rely on SparseGPT as a prior for mask intialization and then compare against SparseGPT for efficacy. This seems to be an unfair comparison, since you're using the method's best result and then improving on top of that. For e.g, without prior masks, the method seems to be much closer in performance to the SparseGPT method for the LLaMA-2 7B model (9.12 vs 10.42). It will be good to see how the method performs for say the 13B model without using any priors.
2. The authors show that SparseGPT has fundamental limits for improvement based on the number of samples (which is well documented and understood) - but the comparisons shown, for e.g., in Figure 4 are with different datasets? Did the authors test the collated datasets for the MaskLLM training with SparseGPT and what plateaus were observed there?
4. One inherent limitation to the method seems that there are many hyper-parameters to tune to find good masks ($\alpha$ for SparseGPT prior, $\mathcal{k}$ for logit scaling and $\tau$ for softmax temperature, $\lambda$ for the regularization coefficient). Do these hyper-parameters scale optimally for all model scales? Or is more fine-grained optimization needed as models scale up?
    - Also for the $\lambda$ parameter, the authors show that using $1e^{-4}$ results in highest gradient norm [Table 12], but the hyper-parameter used in Table 7 is $1e^{-5}$, can the authors clarify this aspect?

**Questions:**

1. For the LLaMA-2 models, the authors mention using the official dataset from the paper. However, the paper has no mention of any datasets used for training the models. Can the authors clarify this discrepancy?
2. In Table 11, when the authors mention the score from RIA, does that include the channel permutation method enabled in the paper?
3. Can the authors verify their magnitude pruning results for downstream tasks for Llama2-7B (Table 1). For such a large perplexity difference between magnitude pruning and SparseGPT / Wanda - the downstream results seem too be high?

**Limitations:**

1. There are some inconsistencies in the links in the paper (for e.g., Appendix D mentioned in page 8 - for weight regularization maps to Section 4.1 in the paper - please fix this).
2. Throughout the paper, it is unclear what results map to what experiments. For example, for Figure 5, which models were used for the ablation? Same for the results in Tables 3, 5, and 13. Scanning the Appendix also did not help clarify this. It makes the results presented somewhat non-trivial to parse.

---

> ### Author Rebuttal · Authors · 2024-08-06
>
> Thank you so much for the invaluable comments and suggestions about baselines, datasets, and hyperparameters!
>
> > **Q1: 1) The comparison between MaskLLM with SparseGPT prior and SparseGPT seems to be an unfair. The method seems to be much closer in performance to the SparseGPT method without prior. 2) It will be good to see how the method performs for say the 13B model without using any priors.**
>
> **A1:** Thank you for the insightful comment. One-shot methods like SparseGPT excel at efficiently identifying a coarse mask. In contrast, MaskLLM, a learning-based method, identifies better masks but is also more resource-intensive. Without a prior, 2,000 steps might not be sufficient to learn all the masks. Moreover, using a trivially-computable Magnitude mask as the prior also yields a respectable PPL of 6.77, compared to the 6.72 achieved with a SparseGPT prior. This is likely due to the high similarities of the masks found by Magnitude and SparseGPT (demonstrated in Figure 8 of the appendix). From our perspective, mask prior is an important design in this work and it's always encouraged and feasible to start training from some prior, which can be easily identified with even magnitude pruning.
>
> **13B model without using any priors:** Regarding the requested 13B experiment, the training is still ongoing. We will keep working to provide new results; however, due to resource and time constraints during the rebuttal period, we cannot immediately present the training results for the 13B model.
>
> > **Q2: The authors show that SparseGPT has fundamental limits for improvement based on the number of samples (which is well documented and understood) - but the comparisons shown, for e.g., in Figure 4 are with different datasets? Did the authors test the collated datasets for the MaskLLM training with SparseGPT and what plateaus were observed there?**
>
> **A2:** For the one-shot method, we use the C4 dataset as was suggested in the original papers. For training with MaskLLM, we utilize a blended dataset collected from the Internet, which covers 69 domains such as programming languages as mentioned in Line 430 of the appendix. Following the advice, we provide more results to evaluate the effectiveness of SparseGPT and MaskLLM on different data sources. The following table shows results obtained by **only** using the blended dataset or the public C4 dataset. It can be observed that the results between datasets are comparable and MaskLLM is still better than the baselines. We will include these additional results in the revised version of the paper.
> |Calibration Data|Blended Data|C4|
> |-|:-:|:-:|
> |Llama-2 7B|5.12|5.12|
> |SparseGPT|9.88|10.42|
> |Wanda|11.25|10.29|
> |MaskLLM (2K steps)|**6.72**|**6.85**|
>
> *Table 1: Wikitext-2 PPL of 2:4 LLaMA-2 7B pruned with different datasets.*
>
> > **Q3: 1) There are many hyper-parameters to tune. Do these hyper-parameters scale optimally for all model scales? Or is more fine-grained optimization needed as models scale up? 2) Why 1e-5 was selected for regularization?**
>
> **A3:** **1)** We agree with the reviewer that our method can be customized by a number of hyperparameters. Fortunately, as demonstrated in Table 7, we applied **the same hyperparameters across all models** and obtained consistently positive results. This indicates that the selected hyperparameters are robust and generalizable to different models and datasets. **2)** Regarding regularization, we opted for the relatively smaller regularization of 1e-5 since it produces a slightly better validation loss (1.83 vs. 1.86) during training. An over-large regularization may limit the searching space of mask learning.
>
> > **Q4: For the LLaMA-2 models, the authors mention using the official dataset from the paper. However, the paper has no mention of any datasets used for training the models. Can the authors clarify this discrepancy?**
>
> **A4:** We do not have access to the official Llama-2 dataset as it is not publicly available. As discussed in Line 431 of the appendix, we collected a blended dataset following similar principles as stated in the Llama paper, encompassing 69 domains such as Programming (C++, C Sharp, Python) and general corpus. We will clarify this point in the revised manuscript. Additionally, we will provide the C4 results mentioned above, and make the code and our learned binary masks public to enhance the reproducibility of our experiments.
>
> > **Q5: In Table 11, when the authors mention the score from RIA, does that include the channel permutation method enabled in the paper?**
>
> **A5:** Thank you for the comment. The RIA result is collected from the official paper. Channel permutation is disabled for this result. If enabled, the PPL of RIA will be 7.77 compared to our 5.85. We will update Table 11 with this result.
>
> > **Q6: Can the authors verify their magnitude pruning results for downstream tasks for Llama2-7B (Table 1). The downstream results seem to be too high?**
>
> **A6:** We double-checked these results and found them to be accurate. The same phenomenon has also been reported in the Wanda paper, where the accuracy drop on zero-shot tasks is 8%.
>
> > **Q7: There are some inconsistencies in the links in the paper (please fix this).**
>
> **A7:** Thank you for the comment. We will fix these links according to the instructions and carefully review other links to ensure their correctness.
>
> > **Q8: Throughout the paper, it is unclear what results map to what experiments. For example, for Figure 5, which models were used for the ablation? Same for the results in Tables 3, 5, and 13.**
>
> **A8:** Tables 3 and 5 utilize GPT-3 2B for quick experiments, while Table 13 covers all three models: GPT-3B, 843M, and Llama-2 7B. We will clarify the models in the captions following your advice.

---

> > ### Comment · Reviewer_2tsd · 2024-08-08
> >
> > Thank you for the detailed response to the reviews and additional experimental results. I understand that the 13B model results will take time, thank your for taking a stab at those.
> >
> > After reading all reviews and responses, I will update my rating to accept (score: 7). Please do incorporate the appropriate fixes / changes in the revised version of the paper.

---

> > > ### Author Response · Authors · 2024-08-08
> > >
> > > Thank you very much for your encouraging feedback! We will ensure that all appropriate fixes and changes are incorporated into the revised version.

---

### Official Review · Reviewer_AwSn · 2024-07-14

**Soundness:** 3
**Presentation:** 4
**Contribution:** 3
**Rating:** 7
**Confidence:** 5

**Summary:**

The authors proposed a novel LLM pruning technique, by modeling the distribution of all possible masks and formulate the selection of optimal masks in a differentiable way.

**Strengths:**

- Important and relevant problem setup.
- The solution is novel.
- Thorough evaluation across a range of model/dataset combinations.

**Weaknesses:**

- Unclear what the computational cost of this technique is, and how that compares with the alternative/more straightforward technique of sparse pretraining/finetuning.
- Unclear why the authors do not report speedup for all models/datasets.
- Using perplexity as a proxy for downstream coding task performance is sub-optimal. Using benchmarks such as HumanEval is preferred.

**Questions:**

- While this work is interesting and thoroughly executed already, can you comment on how this compares with more straightforward baseline of sparse finetuning/pretraining? What's the compute cost v.s. accuracy trade-off?
- Finding 1. is vague in two ways: 1). unclear what "large scale dataset" mean, 1B tokens is not large scale for LLM, please just state the size of the dataset explicitly. 2). I don't think your experiments are enough to justify "Learnable Sparsity ... fully leverage computational resources to learn precise masks through end-to-end training". Just say your technique better leverages computational resources than prior art, which is what your experiments show.
- What's the calibration set size to produce Table 1.?
- For Table.2 SparseGPT, did you also used the weight updates in Learned Mask setup?
- As for weight magnitude regularization (Finding 4), have you tried tuning the learning rate for downstream finetuning? This sounds like a learning rate issue.

**Limitations:**

all addressed.

---

> ### Author Rebuttal · Authors · 2024-08-06
>
> We sincerely thank reviewer AwSn for the valuable comments.
>
> > **Q1: Unclear computational cost. Comparison to sparse pretraining/finetuning.**
>
> **A1:** This submission discusses the computational cost of mask learning in Lines 433-435 of the appendix. According to the original tech report of Llama-2 [1], MaskLLM's training cost is 0.6% of pre-training (1,280 GPU hours / 184,320 GPU hours). On a single node with 8xA6000, MaskLLM (239 s/it) is slightly slower than the Straight Through Estimator (218 s/it) due to its additional sampling process. For a quantitative comparison, please refer to our response to Question 4.
>
> [1] Llama 2: Open Foundation and Fine-Tuned Chat Models
>
> > **Q2: Speedup for all models/datasets.**
>
> **A2:** We supplement more benchmark results for Llama-2 13B and Nemotron-15B here:
> |Model|Input Len.|Output Len.|Dense|2:4|Speed Up|
> |:-|:-:|:-:|:-:|:-:|:-:|
> |**Llama-2 13B**|
> ||128|128|32.97|51.64|1.57×|
> ||128|2048|31.81|49.00|1.54×|
> ||2048|128|31.14|47.19|1.55×|
> ||2048|2048|30.09|45.06|1.50×|
> |**Nemotron-4 15B**|
> ||128|128|38.15|59.40|1.56×|
> ||128|2048|37.51|57.93|1.54×|
> ||2048|128|37.44|57.56|1.54×|
> ||2048|2048|36.85|56.37|1.53×|
>
> *Table 1: Benchmarking Llama-2 13B and Nemotron-4 15B with A6000 and TensorRT-LLM*
>
> The same 2:4 model will exhibit consistent speed-up across different datasets because the acceleration of the 2:4 pattern is determined by the inference engine (TRT-LLM in our case) and the hardware (A6000).
>
> > **Q3: PPL is sub-optimal for coding tasks. Using benchmarks such as HumanEval is preferred.**
>
> **A3:** Thank you so much for the insightful advice. We chose perplexity (PPL) in our paper as it's simple and general across different tasks or domains. We are still working on more specific evaluation metrics, such as HumanEval as suggested by the reviewer, and will provide additional results in the revised appendix.
>
> > **Q4: How this compare with more straightforward baselines of sparse finetuning/pretraining? What's the compute cost v.s. accuracy trade-off?**
>
> **A4:** Following the suggestion, we provide more comparison results for different strategies: (1) Mask-only learning with frozen weights, such as the MaskLLM and Wanda, (2) Sparse Fine-tuning of the remaining weights after pruning and (3) Learning both weight and sparsity, such as STE.
> |Method|Weight Update|LLM|Dense PPL↓|Sparse PPL↓|Δ PPL ↓|HellaSwag ↑|
> |:-|:-:|:-:|:-:|:-:|:-:|:-:|
> |Wanda + Sparse Finetuning [1]|✔️|Llama-1 7B|5.68|7.02|+1.33|n/a|
> |Wanda + LoRA [1]|✔️|Llama-1 7B|5.68|8.24|+2.56|n/a|
> |SR-STE [2]|✔️|Llama-2 7B|5.12|6.80|+1.68|51.26|
> |SparseGPT + SPP [3]|✔️|Llama-2 7B|5.12|n/a|n/a|51.33|
> |Wanda + SPP [3]|✔️|Llama-2 7B|5.12|n/a|n/a|50.61|
> |**MaskLLM + Sparse Finetuning**|✔️|**Llama-2 7B**|**5.12**|**5.83**|**+0.71**|**54.66**|
> |**MaskLLM**|-|**Llama-2 7B**|**5.12**|**6.72**|**+1.60**|**50.91**|
>
> *Table 2: Comparison to more finetuning-based baselines.*
>
> We re-implement SR-STE and collect other results from the related works. Some missing results are marked with "n/a". This table indicates that MaskLLM even without any weight updates, can achieve competitive results (PPl=6.72) compared to SR-STE (PPL=6.80). This is expected as MaskLLM can explore all mask candidates while SR-STE mainly focuses on weights with large magnitudes. Furthermore, incorporating sparse fine-tuning into MaskLLM can significantly improve both perplexity (PPL) and zero-shot accuracy on HellaSwag.
>
> **The cost-accuracy trade-off:** A key message in this work is that incorporating more computing can effectively enhance accuracy. One-shot methods are efficient yet not sufficiently accurate. Techniques like LoRA improve accuracy with more training, but they may not fully explore various mask combinations. In contrast, MaskLLM thoroughly examines different masks through the proposed differentiable sampling, incurring a 9% higher training cost than STE (See Q1), but achieving the highest accuracy among these methods. Practically, one-shot methods are preferable if resources are limited. However, if sufficient resources and samples are available for end-to-end training, MaskLLM is a good choice for compressing LLMs.
>
> [1] A Simple and Effective Pruning Approach for Large Language Models.
> [2] Learning N:M Fine-grained Structured Sparse Neural Networks From Scratch.
> [3] SPP: Sparsity-Preserved Parameter-Efficient Fine-Tuning for Large Language Models.
>
> > **Q5: Finding 1. is vague. 1) just state the size of the dataset explicitly. 2) Just say your technique better leverages computational resources than prior art.**
>
> **A5:** Following the advice, we will replace "large-scale dataset" with "larger calibration set" and specify its exact size (512k samples with 2.097B tokens) in the paper. Regarding point 2, we agree with the reviewer that the phrase "better leverages" is more appropriate and will revise the submission accordingly.
>
> > **Q6: What's the calibration set size to produce Table 1.?**.
>
> **A6:** For the baseline methods, we used 128 samples for one-shot pruning following the official implementation. For MaskLLM, we utilized 512k samples.
>
> > **Q7: For Table.2 SparseGPT, did you also used the weight updates in Learned Mask setup?**.
>
> **A7:** When used as the prior, the weight update in SparseGPT is disabled.
>
>
> > **Q8: As for weight magnitude regularization (Finding 4), have you tried tuning the learning rate for downstream finetuning? This sounds like a learning rate issue.**
>
> **A8:** We indeed conducted tuning experiments and still observed this issue with different learning rates. Our hypothesis is that low magnitude might not be a good initialization for downstream training. The validation PPL at 4,000 training steps is shown bellow, where the regularization effectively alleviates this problem.
>
> |Lr|Val. PPL @ 4K Steps|
> |-|:-:|
> |1e-3|5.91|
> |5e-4|5.85|
> |5e-4 + Reg.|**5.62**|
> |1e-4|5.84|
>
> *Table 3: Validation PPL of downstream finetuning*

---

> > ### Comment · Reviewer_AwSn · 2024-08-07
> > **Acknowledged**
> >
> > Acknowledged and will stick to my accept rating.

---

> > > ### Author Response · Authors · 2024-08-08
> > >
> > > We would like to express our sincere gratitude for the insightful comments! We will improve the quality of our draft following the above suggestions.

---

### Author Rebuttal · Authors · 2024-08-06

We sincerely thank all reviewers for their invaluable comments and suggestions. We will make every effort to provide additional results to support our response within this limited rebuttal period. To ensure better reproducibility, we also promise to release our code and learned masks in the future.

---

### Decision · Program_Chairs · 2024-09-25

**Decision:**

Accept (spotlight)

**Comment:**

The work formulates and applies end-to-end differentiable masking atop pre-trained LLMs and one-shot pruning methods, significantly narrowing the performance gap with dense models with ~1.5x speedups and high parameter-efficiency (size and compute). Masks were also shown as efficient downstream task adapters.

The four reviewers (selected for adjacency to the topic; avg. confidence 4.5) acknowledged the importance, novelty, and thoroughness of the work, ultimately recommending Accept or Strong Accept. The small set of ablative requests were adequately answered. I found the work comprehensive and well-written.

Due to the significant improvement on existing pruning schemes (at ~15B LLM scale, notable for learned methods) and the informative findings and ablations, I believe this LLM PEFT approach and its core concepts will have broad appeal. **I recommend acceptance w/ special presentation.**